# The Effect of TMJ Intervention on Instant Postural Changes and Dystonic Contractions in Patients Diagnosed with Dystonia: A Pilot Study

**DOI:** 10.3390/diagnostics13203177

**Published:** 2023-10-11

**Authors:** Ayse Selenge Akbulut

**Affiliations:** Department of Orthodontics, Peri Sokak Dental Clinic, 42010 Konya, Turkey; selengeakbulut@gmail.com; Tel.: +90-332-503-00-96

**Keywords:** APECS, bruxism, cervical dystonia, dystonia, IDTT, orthodontic, posture, TMD, TMJ

## Abstract

(1) Background: The aim of the present study is to analyze the instant postural changes and changes in the dystonic contractions among patients with dystonia following the use of an intraoral device called a key integrative dynamic TMJ treatment appliance (KIDTA). (2) Methods: Twelve subjects, previously diagnosed with dystonia were enrolled. Their existing records were utilized to assess the changes in their posture and dystonic contractions. The posture analysis was conducted using a mobile application (APECS). The initial records (T0) and records acquired after the delivery of the KIDTA (T1) were utilized in the analysis. The Wilcoxon signed-rank test was performed to compare parameters between T0 and T1, with a significance level set at *p* < 0.05. (3) Results: Based on the Wilcoxon signed-rank test, statistically significant differences in T1 compared to T0 were observed in the severity of dystonic spasms, body alignment, head shift, head tilt, shoulder alignment, shoulder angle, axillae alignment, ribcage tilt, pelvic tilt, knee angle, and tibia angle (*p* < 0.05). (4) Conclusions: Within the limitations of the present pilot study, an intervention to the TMJ through a KIDTA appliance seems to mitigate the severity of dystonic contractions and improve the posture with respect to certain postural parameters.

## 1. Introduction

The temporomandibular joint (TMJ) is a synovial joint consisting of the mandibular condyle and the mandibular fossa, which is located in the temporal bone. An articular disc, a tough fibrous connective tissue composed of compressed collagen fibers, is positioned between the mandibular fossa and mandibular condyle. This articular disc is indispensable for condylar regeneration and may serve as a crucial environmental factor for periosteal activation [1]. The proper functioning of the TMJ and its related components is crucial for directing the movement of the mandible and distributing the stresses that arise from routine activities like chewing, swallowing, talking, yawning, and other functions. On the other hand, parafunctional habits such as clenching, bruxism, pen chewing, nail biting, jaw thrusting, one-sided chewing, and repetitive or continuous external forces that create mandibular deviation in daily life might contribute to the development of temporomandibular disorder (TMD) and related degenerative changes [2,3,4]. Trauma, similar to parafunctional habits, is another factor that causes functional overloading in the TMJ region and contributes to the development of TMD. There is no consensus about the contribution of malocclusion to the development of TMD [5,6,7,8]. Although a previous systematic review referred to the absence of an association between TMD and dental occlusion [5], a high prevalence of TMD was reported in subjects with a dentofacial deformity in a previous meta-analysis [6]. The improper positioning of the mandibular condyle in the articular fossa can create stress in the TMJ complex and, as a result, affect the muscles, fascia, and neural network associated with the TMJ.

On the other hand, dystonia is a neurological condition characterized by involuntary, repetitive, and excessive muscular contractions. These contractions lead to abnormal movements, making it challenging for individuals to control their motions. The resulting movements and incorrect postures can be painful and affect the daily life of the individual. The prevalence of idiopathic or inherited dystonia was reported as 30.85 per 100,000 in a previous meta-analysis [9]. Dystonia was classified based on clinical characteristics and etiological factors by Grütz and Klein [10]. According to the clinical characteristics, dystonia was classified based on age, distribution area in the body, temporal pattern, and associated features. Based on the etiology, dystonia was classified under subcategories such as nervous system pathology, inherited or acquired, and others. Dystonia can impact various parts of the body, such as the neck, face, eyelids, jaw, vocal cords, hands, torso, and feet. Moreover, the repetitive muscle contractions in the affected body parts can lead to a fixed incorrect body posture. Correct posture aims to achieve maximum stability, conserve energy, and minimize stress on anatomical structures, but in patients with dystonia, incorrect posture makes it difficult to achieve these goals [11,12]. The body’s position in space, which is automatically maintained through muscle contractions in response to gravity, can be associated with the TMJ [13,14,15]. Healthy and balanced TMJs are important not only for better chewing ability but also for better posture.

Integrative dynamic TMJ treatment (IDTT) is a treatment protocol developed by the author of the current study aimed at rehabilitating the TMJ by reducing the overloading on the TMJ, releasing probable stress in the relevant neural networks, and providing relaxation in the related muscles in close proximity. This protocol consists of two phases. In the first phase, the aim is to observe whether any overloading in the TMJ exists and whether a problem in the TMJ can be a risk factor for any other disease. These observations are carried out using a key appliance called the key integrative dynamic TMJ appliance (KIDTA) in the initial phase of the treatment protocol. If a positive correlation between the TMJ and symptoms of the disease is identified, the second phase becomes applicable. The second phase comprises multiple active treatment sessions in which integrative medicine techniques and physiotherapy techniques are employed, alongside the utilization of the integrative dynamic TMJ appliance (IDTA).

The aim of the current study is to analyze the acute postural changes and changes in the dystonic contractions in patients with dystonia after the instantaneous repositioning of the TMJ through an intraoral device called the KIDTA. Thus, the study aims to elucidate the effect of increasing the TMJ space on the body posture. The null hypothesis suggests that the immediate repositioning of the TMJ using the KIDTA may not result in alterations in posture and dystonic contractions when observed from both the frontal and lateral perspectives.

## 2. Materials and Methods

The protocol of this retrospective pilot study was approved by the ethical committee of the Department of Medicine at Necmettin Erbakan University (2023/4418). Initial records of subjects referred to Peri Sokak Dental Clinic (Konya, Turkey) between 2022 and 2023 were consecutively recruited.

Inclusion criteria:

Subjects between the ages of 18 and 75;Subjects who have referred to the clinic with complaints of bruxism and/or myofascial pain;Subjects who have received a diagnosis of dystonia by a neurology specialist;Subjects who have started bruxism and TMJ treatment with the first phase of IDTT;Subjects with complete records related to postural assessment before and after intervention.

Exclusion criteria:

Congenital anomaly;Incomplete photograph and video records;Subjects who received botox treatment within the last 4 months;Subjects who started or quit a new medication for the treatment of dystonia within 1 month.

According to the inclusion and exclusion criteria, 12 Caucasian subjects were included in this retrospective archive study.

KIDTA is a soft occlusal appliance made with Functional Impression Tissue Toner (FITT, Kerr, Italy) (Figure 1).

Following the production of the dental models, the transition to the articulator is facilitated through bite registration. The bite registration wax is prepared based on an opening in the anterior region for 3 mm using Fleximeter Strips (Bausch flexi strips, Cologne, Germany). While biting the strips placed between the upper and lower incisors, a slight natural sliding movement in the anterior direction of up to 1 mm is allowed. KIDTA is an appliance that covers both the upper and lower occlusal surfaces. The appliance, which is fabricated using FITT, also covers approximately 1–3 mm of the teeth from the buccal surface. This soft material maintains its form for a duration of up to 10 days, necessitating the potential for periodic redelivery of the KIDTA appliance within the initial phase of treatment.

Postural analysis was performed on previously obtained photos and videos at two timepoints. T0 represented the time before any intervention with KIDTA, while T1 represented the time immediately after the delivery of KIDTA without any long-term usage.

### 2.1. Measurement Method

The photos and videos of the subjects that were acquired before and after intervention were collected. Postural analysis was performed on these photos using an application called APECS-AI Posture Evaluation and Correction System^®^ (APECS mobile application) (New Body Technologies SAS, Grenoble, France). Postural evaluation was conducted in both frontal and lateral views. The landmarks used in postural analysis are provided in Table 1 for frontal view and Table 2 for lateral view.

The images depicting the landmarks are provided in Figure 2 for the frontal view and Figure 3 for the lateral view.

In the frontal view, the vertical reference line was described as a line passing through the midpoint between the MBM points on the right and left sides, perpendicular to the ground surface. In the lateral view, the vertical reference line was determined as the line passing through the LM point and is perpendicular to the ground plane. The horizontal reference line for both planes was described as a line parallel to the ground surface.

The parameters measured in the frontal view and lateral view are detailed in Table 3 and Table 4 for posture analysis, along with their corresponding abbreviations.

The severity of dystonic contraction (SDC) was measured using video records. To calculate the SDC, the number of dystonic contractions per minute was counted. The duration of each individual dystonic contraction was measured in seconds. The measurement of the SDC was achieved by multiplying the total count of dystonic contractions by their respective durations.

Half of the total dataset was re-measured by another investigator to assess inter-rater reliability. To assess inter-rater reliability, an intra-class correlation coefficient was calculated. The intra-class correlation coefficient was above 0.90 which refers to a high reliability between two investigators (Table 5). In the final dataset, only the measurements from the first investigator were utilized.

### 2.2. Statistical Analysis

The statistical analysis of the data was performed using IBM SPSS Statistics Version 26.0 (Chicago, IL, USA). The statistical analysis was carried out through the Wilcoxon signed-rank test for comparing the measurements from before and after the intervention to the TMJ. The Wilcoxon signed-rank test was chosen due to the sample size of 12 participants in the present study, which falls below the conventional threshold of 30 for normality assumptions. According to the guidelines proposed by Sheskin (2003), non-parametric tests like the Wilcoxon signed-rank test are recommended when dealing with small sample sizes, as they can provide reliable inferences without necessitating stringent distributional requirements [16].

## 3. Results

The total sample consisted of eight females and four males, with an age range between 26 and 60 years. The mean values for the age, height, and weight were 41.5, 167.2, and 68.1, respectively. The demographic data of the subjects are provided in Table 6.

Among the 12 subjects, 10 were previously diagnosed with cervical dystonia, 1 subject was diagnosed with oromandibular dystonia, and 1 subject was diagnosed with writer’s cramp.

The descriptive statistics for all parameters are provided in Table 7.

The results of the Wilcoxon signed-rank test are presented in Table 8 for the severity of the dystonic contractions, the postural measurements from the frontal view, and the postural measurements from the lateral view.

The parameter SDC exhibited a statistically significant difference between the measurements of T0 and T1 (*p* < 0.05).

From the frontal view parameters, BAF, HT, SAF, AA, RT, and PTF exhibited statistically significant differences between the measurements of T0 and T1 (*p* < 0.05). However, MIPTT, KAR, KAL, FRR, and FRL showed no statistically significant difference (*p* > 0.05).

Among the lateral view parameters, BAL, HS, SAL, PTL, KA, and TA exhibited statistically significant differences before and after the intervention (*p* < 0.05). However, only FA from the lateral view parameters did not show a statistically significant difference between the two timepoints (*p* > 0.05).

Representative videos of a subject are provided in Appendix A. The alleviation of symptoms associated with dystonic contractions after TMJ intervention can be observed by comparing these before and after videos. Representative photos of two subjects in both the frontal and lateral views showing the alterations in body posture after the TMJ intervention, are presented in Figure 4 and Figure 5.

## 4. Discussion

The results of the current study refer to the changes in certain parameters for both frontal and lateral assessments following intervention in the TMJ area through a KIDTA. These changes encompass not only the alterations in posture but also the alleviation of symptoms associated with dystonic contractions. In view of the findings from the present investigation, it is noteworthy that the null hypothesis was partially rejected.

The current study was designed as a retrospective and pilot study with a small sample size. Dystonia falls within the realm of neurology, and therefore, establishing a study group for this already relatively uncommon disease in the field of dentistry is anticipated to be quite challenging. Moreover, an overview of the information about a new treatment protocol that is under development has been provided in this pilot study. One of the appliances used in this treatment approach was presented with its impact on posture and dystonic contractions.

An incidence of bruxism between 28 and 34% in cervical dystonia and 17–74% in oromandibular dystonia cases was reported in previous studies [17,18,19]. Various kinds of questionnaires are being used for bruxism and TMD evaluation [20,21,22]. However, in the current study, only the subjects who were aware that they had bruxism and myofascial pain, and who were referred to the clinic with one of these chief complaints, were accepted as meeting the criteria for these issues.

Several mobile software applications that utilize artificial intelligence have emerged for the purpose of analyzing and evaluating posture. The PostureScreen^®^ Mobile app is one of the most preferred applications used in previous studies [23,24]. However, extra effort is needed to understand which lines and planes the parameters are formed by. Therefore, the posture analysis was performed through the mobile application called the APECS mobile application in the present study. This application is a reliable tool that also utilizes artificial intelligence [25]. It presents how the relevant parameter is formed with specific points, lines, and planes, solely by analyzing the results. This approach enhances the efficiency of interpreting the analysis results. Moreover, the application provides detailed information about landmarks, contributing to the ease of applicability and simplicity. However, all measurements in this application are expressed as whole numbers. To provide more precise measurements, it is recommended that the program offers values with one or two decimal places after the comma.

The postural analysis and severity of dystonic contractions were evaluated using photos and videos obtained during routine IDTT intervention. Therefore, additional diagnostic tools such as EMG or radiologic images for postural assessment and evaluation of dystonic contractions were not available in this retrospective study. For the evaluation of the changes in dystonic contractions, a new parameter called the ‘severity of dystonic contractions’ was introduced in this pilot study. This parameter was calculated by multiplying the total number of dystonic contractions per minute by the mean value of duration in seconds for all the contractions that appeared in a minute. The maximum total score for the severity of the dystonic contractions was set at 60. Various characteristics of dystonic contractions were observed in the sample. Generally, each contraction lasted for a few seconds, and after a brief resting period, the contraction would start again, and so forth. In these cases, the severity of the dystonic contraction was measured as described, by counting the number of contractions and their duration. However, in some cases, there was only one contraction for the entire minute, which persisted continuously with a fixed head position until the patient intervened with their hands. In such instances, the number of contractions was recorded as one, while the duration was noted as 60 s. The characteristics of the dystonic contractions were also observed as tremors, which manifested as continuous shaking of the head but within a more limited range of motion compared to the previous dystonic contraction types. Due to the brief duration (possibly less than a second) of each tremor contraction and its continuous nature, the severity was determined as 60 for the subjects exhibiting tremors.

Although EMG evaluation would provide more accurate numerical values regarding the contractions, the parameter called the ‘severity of dystonic contractions’ could offer quantitative evaluation of video recordings in cases where EMG is unavailable. On the other hand, the severity of dystonia has previously been described using various scales [26]. Scales such as the Fahn–Marsden Rating Scale, the Bary Albright Dystonia Scale, and the Global Dystonia Severity Rating Scale are some of the scales that have been in use for evaluation of the overall severity of dystonia. These scales encompass the evaluations of affected body parts, triggering factors, related activities, and severity levels, among other factors. However, the severity of the dystonic contractions in the present study involved a simple calculation focusing solely on the contractions’ duration and frequency, without accounting for affected body parts and other factors.

The posture evaluation was conducted using only two views instead of the usual four. The subjects’ heads were positioned to either the left or right side. Many subjects exhibited a body twist and rotation towards one side. In individuals with conditions like kyphosis, lordosis, or sway back, one can anticipate similar postural evaluation outcomes for both left and right lateral assessments. Twisted and/or rotated postures rarely accompany these conditions. However, in subjects with dystonia, a rotated posture is expected to yield contrasting results for the right and left sides of the body during the posture analysis. To address this issue, an additional postural assessment from an axial view is considered necessary. Therefore, postural evaluation from the posterior and left lateral aspects has been reserved for future studies. Such studies would aim to investigate this matter further and utilize a three-dimensional posture analysis to provide a comprehensive explanation.

Various types of occlusal splints, produced through different techniques, have been previously discussed [27]. A previous study reported that both soft and hard occlusal splints have the capacity to alleviate TMJ symptoms [28]. Moreover, during a 4-month follow-up period, soft splints were observed to exhibit superiority over hard splints [28]. Although in some previous studies, the worsening of bruxism by the usage of soft appliances was reported [29,30], the findings of recent studies with objective methodologies offered opposite results [31,32,33]. An increase in the maximum bite force can be associated with bruxism [34]. In a recent study that used a digital gnathodynamometer [31], the maximum bite force was shown to increase when a hard splint was used and decrease when a soft splint was used. In another study where a portable electromyography (EMG) device was utilized, it was concluded that a soft occlusal splint was helpful for relieving pain owing to absorbing occlusal forces [32], whereas hard splints were found to increase muscle pain on palpation. Similarly, an increased bite force was reported for hard occlusal splints compared to soft occlusal splints in a recent study where the measurements were performed through EMG [33]. The most significant characteristic of the KIDTA lies in the material employed during its fabrication. The soft material FITT is fabricated in an articulator rather than a vacuum forming machine. The KIDTA may absorb occlusal forces due to its soft nature, potentially alleviating stress in the TMJ. Moreover, using a soft material is considered as the most proper option when considering the dynamic nature of the overall treatment called IDTT. Rather than determining a proper position of the TMJ initially and keeping that position during treatment, the aim of KIDTA and IDTA is to find the balance of the TMJ during the treatment process of IDTT. Considering the movement of two different joints that are connected to each other, soft appliances could be helpful for the TMJ to find its balance in the treatment process owing to the dynamic changes in the distribution of forces.

The temporomandibular joint (TMJ) is an anatomically important area with close proximity to neural networks. It is richly innervated by sensory nerves, primarily branches of the trigeminal nerve. While the trigeminal nerve itself primarily carries sensory information, its interactions with other cranial nerves can influence motor functions and contribute to various motor disorders. The overload in the TMJ may significantly affect this neural system due to the crucial anatomical connections of the TMJ. By reducing and evenly distributing the load in the TMJ through a soft appliance, an enhancement of the neural system in this region could potentially occur. This mechanism has the potential to provide patients with the opportunity to facilitate their own healing by creating a healthier TMJ environment.

The severity of dystonic contractions decreased after wearing the KIDTA in the current study. Similar mechanisms were explored in other studies [35,36]. Symptom relief for dystonia was reported in a prior study involving the utilization of a hard occlusal stabilization appliance [35]. However, the methodological difference from the current study was the evaluation approach, as questionnaires were employed instead of the video records that were used here. Similarly, in a previous case report, an improvement in motor dysfunction related to Parkinson’s disease (PD) was noted following the combined use of a bite splint alongside oral medication [36]. Another case report documented the enhancement of three subjects with cervical dystonia after utilizing an appliance called an orthotic [37].

In the frontal view, all parameters of the upper body showed a statistically significant difference except for the most intended point of the trunk tilt. However, only the pelvic tilt frontal showed a statistically significant difference among the parameters of the lower body from the frontal view.

A statistically significant decrease in the body alignment from the frontal view was observed in the current study. According to the measurement method in the current study, values closer to 0° refer to a well-aligned body from the frontal view [38]. A decrease in the mean value for the body alignment from 1.42° to 0.08° contributed to a better posture in the present study from the frontal view.

The frontal view parameters about tilts were measured through the lines formed by the connection of the bilateral identical points. The lobulus auriculae was the reference point for assessing the head tilt. A similar horizontal level of the lobulus auriculae points indicates a normal alignment [38]. The initial mean value of the head tilt was 9.92°, indicating a severe head tilt in patients with dystonia. However, this mean value decreased to 1.25° after the intervention, approaching the normal range of 0–0.1° [38]. Our results for the head tilt were also compatible with the mean value of 2° reported in a previous study conducted on young, healthy individuals [39].

The shoulder alignment in the frontal view was measured based on the acromion point in the current study. The normal value for the shoulder alignment was reported as 181°, which translates to 1° based on the current measurement method in a previous study [38]. The only difference was the reference point for the measurement, which was the coracoid process in the previous study. When assessing the tilt in the frontal view, symmetry is of the utmost importance. Therefore, even when different reference points were used in the measurements, similar norms could apply to the measurements performed based on close proximity. Considering a normal value of up to 1°, the mean value of 2.83° also indicates a tilt in the shoulder in subjects with dystonia. However, the decrease of this value to 0.33° after the intervention reveals a correction in the shoulders as well. A shoulder alignment of 1.3° in a previous study was also in line with the current results for T1 [39].

A significant decrease in the axillae alignment, ribcage tilt, and pelvic tilt after the use of the KIDTA indicates a correction in these postural parameters. The mean values for these parameters (AA: 0.58°, RT: 0.83°, PT: 0.83°) in T1 were also consistent with the mean values of the previous study (AA: 1.3°, RT: 1.9°, PT: 2.3°) that utilized the same measurement method [39].

The trunk inclination was reported as 1.6° in a previous study in which young healthy adults were examined [39]. This result was consistent with the findings for the most intended point of the trunk tilt in both T0 and T1 parameters (MIPTT-Pre: 0.83°, MIPTT-Post: 0.42°). This similarity could explain the non-significant difference between them. Similarly, the frontal parameters concerning the knee angle did not show a statistically significant difference between T0 and T1, with mean values ranging between 6.75° and 6.50°. These values were also similar to those in a previous study that used the same landmarks [39].

All parameters showed a statistically significant difference in the lateral view parameters except for the foot angle. When compared to T0, the body alignment lateral, head shift, pelvic tilt, knee angle, and tibia angle were decreased in T1 in the lateral view. However, the shoulder angle increased in T1 compared to T0.

An increased value of the body alignment lateral in subjects with dystonia was observed in the present study (mean: 3.58°). However, intervention to the TMJ enabled the subjects to maintain a better posture from the lateral view with a better body alignment (mean: 1.00°).

The head shift was measured as the angle between the vertical reference line and the line connecting the C7 and TG points in the present study. A similar measurement was performed in previous studies as the angle between the horizontal line and the line connecting the tragus with C7 [38]. This measurement was named as the craniovertebral angle. An increase in the craniovertebral angle means a less forward head posture [40]. Therefore, the decreased values of the head shift in T1 (mean: 36.17°) compared to T0 (mean: 49.75°) indicate an enhancement in forward head posture in the present study. However, the head shift after the intervention was less than 31.4, which was reported in a previous study [38]. This can be explained by the study sample of the previous study, which consisted of healthy subjects. In another previous study, the mean craniovertebral angles ranged between 32.67° and 46.83°. These values referred to a more forward head posture compared to the T1 values. However, they were compatible with the results in T0. The mean age could explain this, with a mean age of 78.42 in the previous study and a mean age of 41.5 in the current study [41].

The measurement method of the shoulder angle was similar to the studies in the literature [38,40]. Lower values of the shoulder angle indicate a more kyphotic posture, forward head posture, and rounded shoulders [40]. The shoulder angle was increased in T1 (mean: 39.42°) compared to T0 (mean: 23.83°). This indicated an improvement in the shoulder angle. However, the values even after the intervention could not reach the shoulder angles of the other studies. Mean values of 51.4° and 53.7° were reported in previous studies where asymptomatic subjects were examined [38,42]. However, the results were higher than a previous study where a mean angle of 19.6° was reported [39]. According to the results of that previous study, the shoulder alignment was the only parameter in sagittal view that failed the reproducibility analysis.

In a previous study, it was suggested that an anterior pelvic tilt up to a certain degree could be a typical finding in asymptomatic subjects [43]. They reported mean degrees of the anterior pelvic tilt as 6.74° and 6.23° for the left and right sides, respectively, in males. For females, the values were reported as 6.93° for the left side and 6.63° for the right side [43]. In another study conducted on healthy adults and athletes, the mean values of the anterior pelvic tilt were reported as 9.6° and 11.7° for males and females, respectively [44]. Furthermore, in a study based on radiographic images, a mean anterior pelvic tilt value of 13° was reported [45]. In another study conducted on healthy college students, the mean anterior pelvic tilt values were reported as 8.6° (right) and 8.7° (left) for males, and 12.2° (right) and 11.8° (left) for females [46]. In a previous study, a postural assessment was conducted through a mobile application from the sagittal view [39]. The mean value of the pelvic tilt was reported as 16.9° in young healthy adults. A statistically significant decrease in the mean values of the pelvic tilt from the lateral view was observed in the current study after the intervention. The mean values for T0 (13.25°) and T1 (11.75°) could be evaluated as consistent with the previous studies [44,45,46]. However, the lack of consensus about the norms of pelvic tilt should be considered when evaluating the pelvic tilt.

A significant difference in the knee angle and tibia angle was also observed in the current study. Although the intervention did not affect the lower limbs in the frontal view, it caused a change in the legs when evaluated from the lateral view. This effect could be related to changes in the pelvic tilt as well as the enhancement of the overall posture in the lateral view. The relationship between the anterior pelvic tilt and lower extremity kinematics was presented in a previous study [47]. Although static posture was investigated in the current study, the association between the legs and the pelvic tilt could also have an impact on static posture as well.

The total sample size was the main limitation of the current study, which was conducted on 12 subjects. Due to the relative rarity of dystonia as a neurological disorder, obtaining a large sample size in the dental area is challenging. Therefore, the study was designed as a pilot study. Another limitation was related to the selection criteria. The subjects were not using the same medications. This could affect the results. It is believed that changing or discontinuing the medications that patients routinely use may have some effects on the posture and dystonic contractions. To avoid influencing the parameters due to any changes in medication, the medications were not altered, and the comparisons were performed between before and after the intervention within groups. For further studies, a sample that includes individuals undergoing the same medical interventions could be considered to eliminate the effects of factors other than the TMJ intervention.

The current pilot study presented the immediate changes that occurred in the static posture and dystonic contractions after intervention to the TMJ. An overview of a new therapeutic approach under development was also provided in the current study. The findings emphasize the importance of the TMJ not only for healthy individuals but also for patients with dystonia. By presenting the relationship between the TMJ and posture in the current study, the importance of the TMJ has once again been highlighted. Clinical suggestions and implementations of the current study could include taking preventative measures to maintain TMJ health, conducting detailed routine examinations of the TMJ, treating temporomandibular disorders with appropriate timing and methods, recommending the inclusion of routine TMJ evaluations in the assessment process for individuals with dystonia, and being open to multidisciplinary collaborations that consider a holistic approach to the human body.

## 5. Conclusions

Within the limitations of the present pilot study, an intervention in the TMJ position through the KIDTA can lead to immediate changes in the posture of subjects with dystonia, including the body alignment, head shift, head tilt, shoulder alignment, shoulder angle, axillae alignment, ribcage tilt, pelvic tilt, knee angle, and tibia angle. Additionally, this intervention could mitigate the severity of dystonic spasms in subjects with dystonia.

## Figures and Tables

**Figure 1 diagnostics-13-03177-f001:**
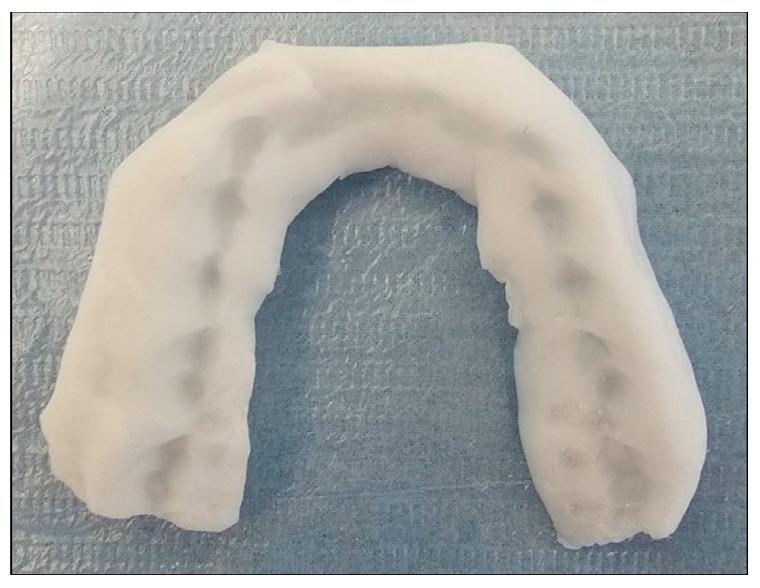
Key Integrative Dynamic Temporomandibular Joint Appliance (KIDTA).

**Figure 2 diagnostics-13-03177-f002:**
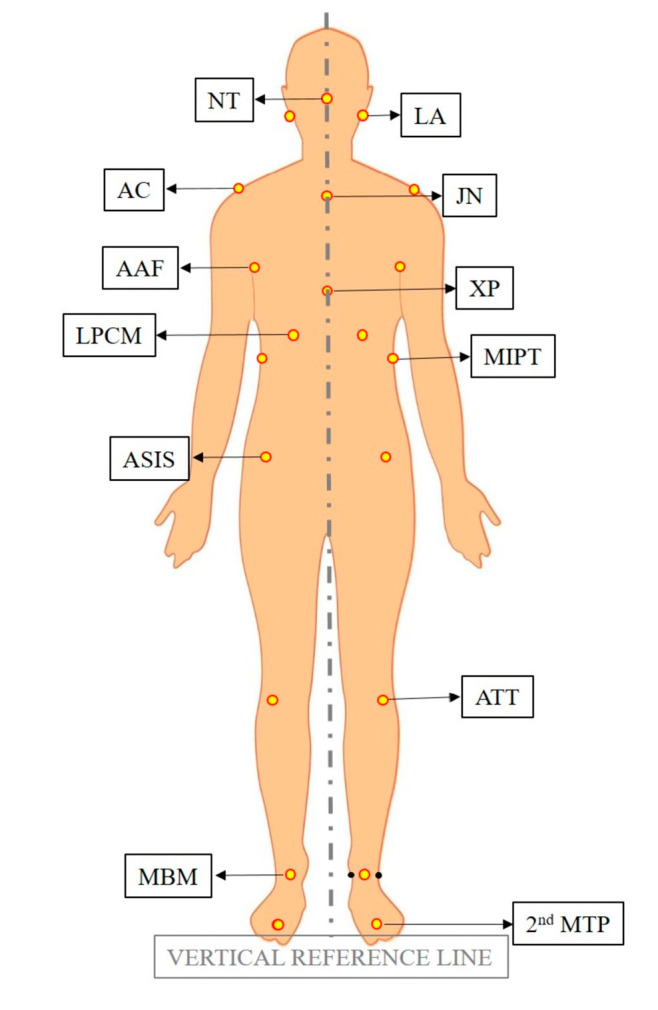
Landmarks used in frontal view.

**Figure 3 diagnostics-13-03177-f003:**
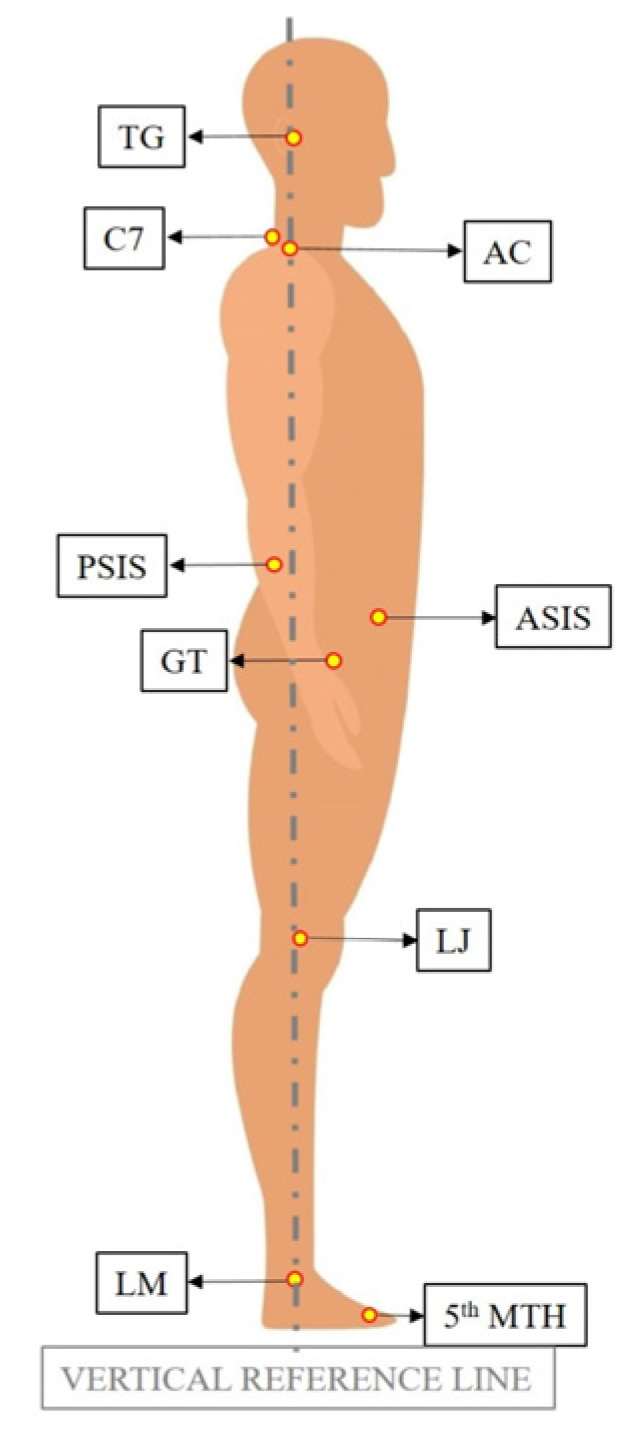
Landmarks used in lateral view.

**Figure 4 diagnostics-13-03177-f004:**
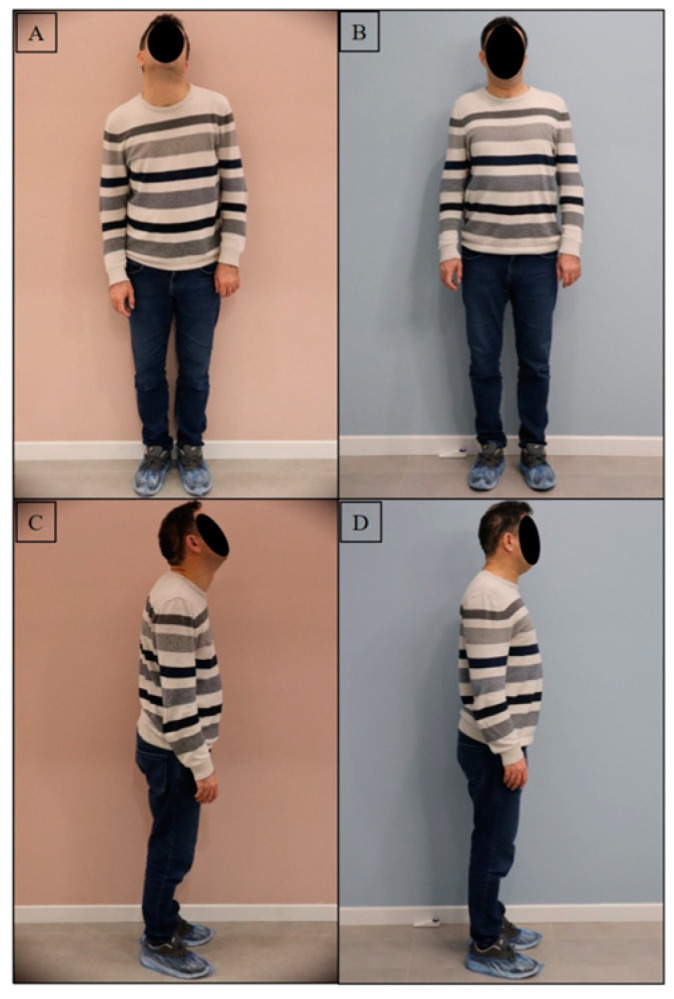
A representative photo of a subject presenting the alterations in body posture after TMJ intervention. (**A**) Posture in frontal view before TMJ intervention at T0, (**B**) posture in frontal view after TMJ intervention at T1, (**C**) posture in lateral view before TMJ intervention at T0, (**D**) posture in lateral view after TMJ intervention at T1.

**Figure 5 diagnostics-13-03177-f005:**
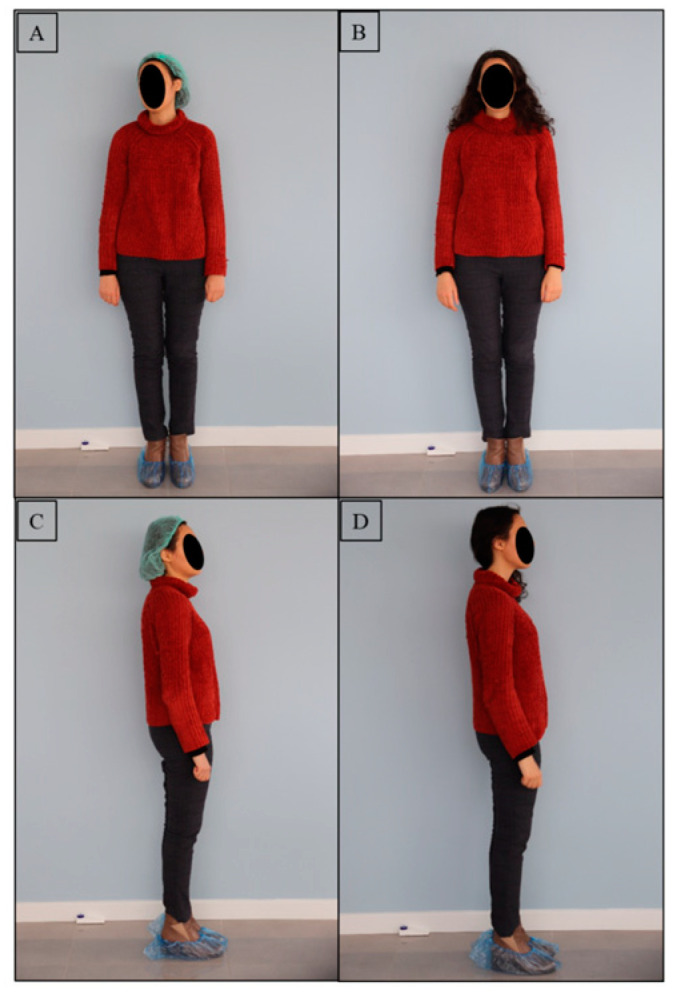
A representative photo of a subject presenting the alterations in body posture after TMJ intervention. (**A**) Posture in frontal view before TMJ intervention at T0, (**B**) posture in frontal view after TMJ intervention at T1, (**C**) posture in lateral view before TMJ intervention at T0, (**D**) posture in lateral view after TMJ intervention at T1.

**Table 1 diagnostics-13-03177-t001:** Descriptions and abbreviations of landmarks used in frontal view.

Landmark	Abbreviation	Description
Lobulus auriculae	LA	The lowest part of the ear.
Nose tip	NT	The tip of the nose.
Acromion	AC	The outer end of the scapula, extending over the shoulder joint and forming the highest point of the shoulder to which the clavicle is attached.
Jugular notch	JN	The largest visible dip in between the neck and the two clavicles in the superior margin of the sternum.
Anterior axillary fold	AAF	The ridge of the skin-covered inferior border of the pectoralis major muscle along the anterior sides of the chest where the underside of each arm meets the shoulder.
Xiphoid process	XP	The pointed process of the cartilage at the lowest part of the sternum.
Lowest point of costal margin	LPCM	The most intended point on the contour of the torso inferiorly.
Most intended point of the trunk	MIPT	The most intended point on the contour of the torso laterally.
ASIS	ASIS	The prominence at the anterior projection of the iliac crest.
Anterior tibial tuberosity	ATT	The large oblong elevation on the proximal anterior aspect of the tibia.
Midpoint between malleoli	MBM	A midpoint of the line joining the most prominent bone on the outer side of the ankle and the most prominent bone on the inner side of the ankle.
Second metatarsophalangeal joint	2nd MTP	The joint between the metatarsal bone of the foot and the second proximal phalange.

**Table 2 diagnostics-13-03177-t002:** Descriptions and abbreviations of landmarks used in lateral view.

Landmark	Abbreviation	Description
Tragus	TG	A small pointed prominence of the external ear, situated in front of the ear canal.
C7 vertebra	C7	The most visible protrusion at the base of the neck.
Acromion	AC	The outer end of the scapula, extending over the shoulder joint and forming the highest point of the shoulder, to which the clavicle is attached.
ASIS	ASIS	The prominence at the anterior projection of the iliac crest.
PSIS	PSIS	The dimples between the buttocks and waist.
Greater trochanter	GT	A large protrusion located toward the top part of the shaft of the femur, facing outward.
Lateral joint	LJ	A slightly rounded enlargement on the surface of the knee.
Lateral malleolus	LM	The most prominent bone on the outer side of the ankle.
Head of the 5th metatarsal bone	5th MTH	A point at the prominence of the long bone (palpable along the distal outer edges of the feet) that connects the fifth proximal phalange to tarsal bone.

**Table 3 diagnostics-13-03177-t003:** Parameters measured in frontal view.

Parameter	Abbreviation	Description
Body Alignment Frontal	BAF	The angle formed between the vertical reference line and the line passing through the JN and the midpoint between the MBM points on the right and left sides.
Head Tilt	HT	The angle formed by the horizontal reference plane and the line joining the LA right and the LA left.
Shoulder Alignment	SAF	The angle formed by the horizontal reference plane and the line joining the AC right and the AC left.
Axillae Alignment	AA	The angle formed by the horizontal reference plane and the line joining the AAF right and the AAF left.
Ribcage Tilt	RT	The angle formed by the horizontal reference plane and the line joining the LPCM right and the LPCM left.
Most intended point of the trunk Tilt	MIPTT	The angle formed by the horizontal reference plane and the line joining the MIPT right and the MIIPT left.
Pelvic Tilt Frontal	PTF	The angle formed by the horizontal reference plane and the line joining the ASIS right and the ASIS left.
Right Knee Angle	KAR	The acute angle formed between the line connecting the ASIS right and the ATT right, as well as the line connecting the ATT right and the MBM right.
Left Knee Angle	KAL	The acute angle formed between the line connecting the ASIS left and the ATT left, as well as the line connecting the ATT left and the MBM left.
Right Foot Rotation	FRR	The angle between the vertical reference line passing through the MBM right and the line connecting the MBM right and the 2nd MTP right.
Left Foot Rotation	FRL	The angle between the vertical reference line passing through the MBM left and the line connecting the MBM left and the 2nd MTP left.

**Table 4 diagnostics-13-03177-t004:** Parameters measured in lateral view.

Parameter	Abbreviation	Description
Body Alignment Lateral	BAL	The angle between the vertical reference line and the line connecting the LM and TG points.
Head Shift	HS	The angle between the vertical reference line and the line connecting the C7 and TG points.
Shoulder Angle	SAL	The angle between the horizontal reference line and the line connecting the C7 and AC points..
Pelvic Tilt Lateral	PTL	The acute angle between the horizontal reference line and the line connecting the ASIS and PSIS points.
Knee Angle	KA	The acute angle between the vertical reference line and the line connecting the GT and LJ points.
Tibia Angle	TA	The acute angle between the vertical reference line and the line connecting the LJ and LM points.
Foot Angle	FA	The angle between the horizontal reference line and the line connecting the LM and 5th MTH points.

**Table 5 diagnostics-13-03177-t005:** Inter-examiner reliability.

	Intra-Class Correlation	95% Confidence Interval		F Test with True Value 0	
		Lower Bound	Upper Bound	Value	df1
Average Measures	90.4%	0.860	0.935	10.434	107

**Table 6 diagnostics-13-03177-t006:** Demographic Data.

	*N*	%	Age			Height		Weight	
			Range	Mean	S.D	Mean	S.D.	Mean	S.D.
Female	8	66.67%	26–60	42.8	12.8	164	4.5	66.0	13.2
Male	4	33.33%	26–49	39	10	173.5	2.6	72.3	6.1
Total	12	100%	26–60	41.5	11.6	167.2	6	68.1	11.4

S.D. Standard Deviation.

**Table 7 diagnostics-13-03177-t007:** Descriptive statistics for all parameters.

	*N*	Range	Minimum	Maximum	Mean		S.D.
					Statistic	Std. Error	Statistic
Severity of Dystonic Contraction-SDC-Pre	12	60.00	0.00	60.00	45.67	5.47	18.97
Severity of Dystonic Contraction-SDC-Post	12	26.00	0.00	26.00	12.17	2.53	8.78
Body Alignment Frontal-BAF-Pre	12	2.00	1.00	3.00	1.42	0.19	0.67
Body Alignment Frontal-BAF-Post	12	1.00	0.00	1.00	0.08	0.08	0.29
Head Tilt-HT-Pre	12	24.00	1.00	25.00	9.92	2.04	7.06
Head Tilt-HT-Post	12	4.00	0.00	4.00	1.25	0.41	1.42
Shoulder Alignment-SAF-Pre	12	9.00	0.00	9.00	2.83	0.73	2.52
Shoulder Alignment-SAF-Post	12	2.00	0.00	2.00	0.33	0.19	0.65
Axillae Alignment-AA-Pre	12	10.00	0.00	10.00	2.75	0.84	2.90
Axillae Alignment-AA-Post	12	2.00	0.00	2.00	0.58	0.19	0.67
Ribcage Tilt-RT-Pre	12	6.00	0.00	6.00	1.50	0.49	1.68
Ribcage Tilt-RT-Post	12	3.00	0.00	3.00	0.83	0.32	1.12
Most intended point of the trunk Tilt-MIPTT-Pre	12	3.00	0.00	3.00	0.83	0.32	1.12
Most intended point of the trunk Tilt-MIPTT-Post	12	1.00	0.00	1.00	0.42	0.15	0.52
Pelvic Tilt Frontal-PTF-Pre	12	5.00	0.00	5.00	2.33	0.50	1.72
Pelvic Tilt Frontal-PTF-Post	12	2.00	0.00	2.00	0.83	0.24	0.84
Right Knee Angle-KAR-Pre	12	11.00	1.00	12.00	6.75	0.85	2.96
Right Knee Angle-KAR-Post	12	11.00	1.00	12.00	6.67	0.87	3.03
Left Knee Angle-KAL-Pre	12	11.00	3.00	14.00	6.75	1.03	3.57
Left Knee Angle-KAL-Post	12	10.00	3.00	13.00	6.50	0.10	3.45
Right Foot Rotation-FRR-Pre	12	36.00	−8.00	28.00	5.75	2.44	8.43
Right Foot Rotation-FRR-Post	12	21.00	0.00	21.00	5.92	1.60	5.53
Left Foot Rotation-FRL-Pre	12	25.00	−7.00	18.00	5.92	2.15	7.44
Left Foot Rotation-FRL-Post	12	21.00	0.00	21.00	6.25	1.89	6.54
Body Alignment Lateral-BAL-Pre	12	7.00	1.00	8.00	3.58	0.65	2.23
Body Alignment Lateral-BAL-Post	12	3.00	0.00	3.00	1.00	0.30	1.05
Head Shift-HS-Pre	12	80.00	29.00	109.00	49.75	6.18	21.41
Head Shift-HS-Post	12	26.00	21.00	47.00	36.17	2.06	7.13
Shoulder Angle-SAL-Pre	12	53.00	6.00	59.00	23.83	4.40	15.23
Shoulder Angle-SAL-Post	12	47.00	18.00	65.00	39.42	4.00	13.85
Pelvic Tilt Lateral-PTL-Pre	12	12.00	9.00	21.00	13.25	1.03	3.57
Pelvic Tilt Lateral-PTL-Post	12	11.00	6.00	17.00	11.75	0.91	3.14
Knee Angle-KA-Pre	12	6.00	0.00	6.00	3.42	0.57	1.98
Knee Angle-KA-Post	12	5.00	0.00	5.00	2.92	0.48	1.68
Tibia Angle-TA-Pre	12	10.00	1.00	11.00	5.75	0.83	2.86
Tibia Angle-TA-Post	12	9.00	0.00	9.00	4.75	0.77	2.67
Foot Angle-FA-Pre	12	22.00	16.00	38.00	28.25	1.92	6.65
Foot Angle-FA-Post	12	18.00	17.00	35.00	27.58	1.75	6.05

S.D.: Standard Deviation. Pre: measurement before intervention in T0. Post: measurement after intervention T1.

**Table 8 diagnostics-13-03177-t008:** The comparison of the parameters through the Wilcoxon signed-rank test.

		Total *N*	Test Statistic	Standard Error	Standardized Test Statistic	Asymptotic Sig. (2-Sided Test)
	Severity of Dystonic Contraction-SDC	12	0.000	11.247	−2.934	0.003 *
**Frontal View**	Body Alignment Frontal-BAF	12	0.000	10.909	−3.025	0.002 *
	Head Tilt-HT	12	0.000	11.242	−2.936	0.003 *
	Shoulder Alignment-SAF	12	0.000	9.753	−2.820	0.005 *
	Axillae Alignment-AA	12	2.000	9.753	−2.615	0.009 *
	Ribcage Tilt-RT	12	0.000	4.500	−2.333	0.020 *
	Most intended point of the trunk Tilt-MIPTT	12	2.000	3.623	−1.518	0.129
	Pelvic Tilt Frontal-PTF	12	0.000	9.657	−2.848	0.004 *
	Right Knee Angle-KAR	12	4.000	2.646	−0.378	0.705
	Left Knee Angle-KAL	12	0.000	1.732	−1.732	0.083
	Right Foot Rotation-FRR	12	4.000	2.739	−0.365	0.715
	Left Foot Rotation-FRL	12	13.500	7.045	−0.639	0.523
**Lateral View**	Body Alignment Lateral-BAL	12	0.000	11.164	−2.956	0.003 *
	Head Shift-HS	12	0.000	12.723	−3.065	0.002 *
	Shoulder Angle-SAL	12	66.000	11.247	2.934	0.003 *
	Pelvic Tilt Lateral-PTL	12	6.000	9.657	−2.226	0.026 *
	Knee Angle-KA	12	0.000	4.287	−2.449	0.014 *
	Tibia Angle-TA	12	0.000	8.147	−2.762	0.006 *
	Foot Angle-FA	12	8.000	8.016	−1.809	0.070

* Statistically significant difference (*p* < 0.05).

## Data Availability

Research data are available on request due to privacy or ethical restrictions.

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
