# Peer review of "The Effect of TMJ Intervention on Instant Postural Changes and Dystonic Contractions in Patients Diagnosed with Dystonia: A Pilot Study"

_diagnostics, 2023, doi:10.3390/diagnostics13203177_

Round 1

Reviewer 1 Report

A study on an interesting topic, hard work, documented. The discussions correctly show the limitations of the study, but the general conclusion shows that still identifying patients with Dystonia and starting the treatment will help the posture. From my point of view, I think that the study meets the scientific criteria to be published.

Author Response

Dear reviewer,

Thank you for your valuable time and for your insightful comments about the article. The author greatly appreciates your recognition of the limitations of our study, which indeed revolved around the relatively small sample size of 12 subjects. The author completely acknowledges that, given the rare occurrence of dystonia disease as a neurological condition within the dental field, assembling a more extensive sample proved to be a significant challenge. Consequently, the study was intentionally structured as a pilot investigation. Thank you for your understanding of this aspect of our research. The author would like to express special thanks for your encouragement and contributions.

Kindest Regards,

Reviewer 2 Report

The study is very interesting and has major importance for Dystonia patients. I have a few comments that I believe will improve the study design.

In Line 36 it says that malloclusion is an etiological factor for TMD. However, there are several studies showing that malloclusion does not cause TMD. Please clarify that in the introduction section. 

The Key Integrative Dynamic TMJ Appliance is a soft appliance. There are studies showing that soft appliance worsen bruxism. Should it be considered the best option for Dystonia patients?

I encourage the results of Intra and interkappa to be describe in the methods section.

Also, I would suggest that Tables would show the percentages of the results.

The study is well written!
